# Cross-Cultural Adaptation and Psychometric Characteristics of the Greek Functional Gait Assessment Scale in Healthy Community-Dwelling Older Adults

Sofia Lampropoulou [1,2,*], Anthi Kellari [1,3], Ingrid A. Gedikoglou [4], Danai Gagara Kozonaki [1], Polymnia Nika [1] and Vasiliki Sakellari [5]

[1] Physiotherapy Program, Life and Health Sciences Department, University of Nicosia, P.O. Box 24005, Nicosia CY-1700, Cyprus; akellari@uth.gr (A.K.); dgkozonaki@gmail.com (D.G.K.); polymnia.nika@gmail.com (P.N.)
[2] Physiotherapy Department, School of Health Rehabilitation Sciences, University of Patras, University Campus, 26504 Rio, Greece
[3] Physiotherapy Department, School of Health Sciences, University of Thessaly, 35100 Lamia, Greece
[4] Physiopoint, 16674 Athens, Greece; inged.physio@gmail.com
[5] Physiotherapy Department, Faculty of Health and Care Sciences, University of West Attica, Agiou Spiridonos 28, Egaleo, 12243 Athens, Greece; vsakellari@uniwa.gr
[*] Correspondence: lampropoulou@upatras.gr; Tel.: +30-2610-962410

**Abstract:** The Functional Gait Assessment (FGA) was cross-culturally adapted into Greek, according to international guidelines. The final Greek version of the scale ($FGA_{GR}$) was evaluated for its reliability and was correlated with the mini-Balance Evaluation Systems Test (mini-BESTest), the Berg Balance Scale (BBS), the Timed Up and Go (TUG) test, and the Falls Efficacy Scale-International (FES-I) questionnaire, for testing the concurrent validity. The discriminant validity between individuals reporting low and those reporting high concern about falls as well as the predictive validity in identifying people with high risk of falls were assessed. The $FGA_{GR}$ was characterized as comprehensible in its content and orders. Psychometric testing in 24 Greek-speaking individuals (six men and eighteen women, $66 \pm 7$ years old) yielded excellent test-retest (ICC = 0.976) and inter-rater reliability (ICC = 0.984), but moderate internal consistency (Cronbach's alpha = 0.660). The $FGA_{GR}$ scale proved its concurrent and discriminant validity while a maximum cutoff point of 25, with sensitivity of 84% and specificity of 100%, was identified to be optimal for predicting risk of falls in the elderly. The good psychometric characteristics of the $FGA_{GR}$ confirm its applicability in assessing gait of Greek-speaking older adults.

**Keywords:** FGA; validity; reliability; elderly; gait; balance

## 1. Introduction

Human gait is fundamentally complex and it requires the cooperation of many structures and systems to be organized and successfully executed according to the demands of different activities and environments [1]. Voluntary movements, automatic responses, cognitive readjustments or emotional motor behaviors during gait are elicited by the interconnection between the higher executive centers of the brain such as the brainstem, the cerebellum, the basal ganglia, and the motor cortex with pathways and with the spinal cord [2,3]. Their interconnection with the limbic system and hypothalamus mediates the emotional motor behavior [4].

Deviations from the normal pattern of gait and mobility are very common, as a consequence of advanced age, neurological and musculoskeletal injuries, sensory deficits, and neurodegenerative diseases [5,6]. Abnormal gait is a common sign in the elderly with a prevalence of 35%, which increases to 60% over 80 years [7]. Among various gait disturbances, which are observed in the elderly, the most important of them include

the decrease in speed, the increase in the double-stance phase time, the widening of the support base, and the decrease in stride length [5]. Additional changes are evident in spatiotemporal gait parameters, with the most eminent being the unsteadiness in the dark and on uneven surfaces, and the impaired dual-tasking ability while walking [6]. The aforementioned implications indicate the importance and necessity of a correct and comprehensive assessment of gait and balance in the elderly. By identifying deficits and organizing a treatment plan, falls might be prevented and the elderly would maintain their autonomy, their functionality, and their quality of life [8,9].

A variety of assessment methods for gait and balance are available [10–12]. The functional scales have the advantage of being easily applicable in the clinical environment and providing reliable and valid records of the ability of the person to perform simple daily activities [13]. The FGA has been specifically developed by Wrisley et al. as a modified form of the DGI in order to assess balance during gait in people with vestibular disorders [14]. Since then, it has been extensively used in other populations, such as in older adults [15,16], in Parkinson's disease [17], in multiple sclerosis [18], in stroke [19], and in incomplete spinal cord injury [20]. It is a 10-item scale which assesses gait in ten different functional conditions: with normal and alterations in speed, under vertical and horizontal head turns, over obstacles, with pivot turns, on stairs, in a narrow base of support, with eyes closed and walking backwards [14,21,22]. The application of the scale takes only 10 min, and it requires very simple and easy to find equipment, such as two shoe boxes, some steps, and a free 6 m walkway of 30.48 cm width, marked at its beginning and end [14]. The original FGA has presented high inter- and intra-rater reliability as well as internal consistency [14]. Its relation to other balance and gait tests, such as the Activities-Specific Balance Confidence Scale (ABC), Dizziness Handicap Inventory (DHI), Perception of Dizziness Symptoms (PDS), Timed Up and Go (TUG) test, and Dynamic Gait Index (DGI), was moderate [14,22]. The FGA has presented its discriminant and predictive validity in classifying risk of falls in older adults and predicting falls in community-dwelling older adults at a cutoff score of 22/30 [16].

Adaptations of the scale are available in German, Persian, and Brazilian, which have yielded high intra- and inter-rater reliability, significant correlations with other balance and gait measures, and good internal consistency [23–25]. The Brazilian version revealed an ability to discriminate between older adults with low and high concern about falls and a cutoff score of 22/30 points for predicting falls in community-dwelling older adults [26]. No other adapted versions have been evaluated for their discriminant and predictive validity. So far, a functional scale that can assess balance deficits during walking is not available in Greek. The literature indicates good psychometric characteristics of the scale and easy applicability in the clinical environment. In addition, its ability to predict risk for falls is an extremely useful indicator for clinicians in order to design therapeutic protocols for prevention of falls. Thus, its adaptation into Greek would be necessary to facilitate the assessment of balance during walking in Greek-speaking older adults who live in the community. With the translation of the FGA scale, there will be in Greek for the first time a scale that purely assesses balance during walking. The purpose, therefore, of this research is two-fold: (i) the translation and cultural adaptation of the FGA functional gait assessment scale into Greek, and (ii) its psychometric testing in elderly people.

## 2. Materials and Methods

After taking permission from the developers of the FGA scale (Dr. DM Wrisley), this study was completed in two parts. The first part (I) comprised the cross-cultural adaptation process, and in the second part (II) the psychometric test of the adapted scale was completed.

### 2.1. (I) Cross-Cultural Adaptation

The cross-cultural adaptation process was divided into five stages and was undertaken based on international guidelines for cross-cultural adaptation of health care scales [27].

Starting the process, two native Greek translators individually translated the scale from English to Greek (*forward translation stage*), and the two separate translations were then synthesized into one common translation (*synthesis I stage*). In the third stage, the synthesized Greek scale was translated back into English (*backward translation stage*) by two other translators, and the two resulting English translations were then synthesized to produce an English version (*synthesis II stage*). This synthesis was completed after comparisons with the original English scale for the most correct syntax and relevance of the content. In addition, there was a collaboration with the authors of the original version of the FGA, which helped to solve questions and formulate more precisely the statements of the activities, to finally produce the pre-final form of the Greek FGA (pre-final $FGA_{GR}$). In the fifth and last stage, the pre-final $FGA_{GR}$ was distributed to a sample of physiotherapists to pilot test the clarity and comprehensibility of the translated content. Since no further changes were required, this final version of the $FGA_{GR}$ was further tested for its psychometric characteristics in a sample of healthy older adults (see the next section of "$FGA_{GR}$ Psychometric Testing").

The translation of the FGA into the Greek language was achieved with the collaboration of four translators. All translators had an excellent knowledge of both English and Greek language since they had lived in England for many years or were English citizens with Greek parents. At every stage of translation (forward or backward), one of the translators was a physiotherapist and the other one had a deep knowledge of the cultural context and linguistic nuances of the target language. All translators had the instruction to perform a conceptual rather than a literal translation and to keep notes and comments about the easiness of the translation process. They were allowed to use translation instruments or dictionaries, but they had to be blinded to the original English instrument during the back translation (from Greek to English). The committee of the translators was in close collaboration with the head of the adaptation process (SL) who also served as a third independent translator.

### 2.2. (II) $FGA_{GR}$ Psychometric Testing

#### 2.2.1. Sample

The pre-final version of the $FGA_{GR}$ was distributed to 6 new physiotherapists (1–5 years of experience, with low risk to the habitual use of the scale due to experience), to test the clarity of the content and the comprehensibility of the commands and instructions.

In addition, the final adapted scale was undertaken in a convenience sample of healthy male and female older adults, recruited from a well-known multidisciplinary center in Nicosia, Cyprus and through acquaintances of the physical therapy students at the University of Nicosia. All participants had to be over 65 years and walk independently and without a walking aid. Although the FGA scale can also be applied to people who use a walking aid, the ability to move without an aid was chosen as a necessary condition for the homogeneity of the sample. Participants should not have mental or cognitive problems, in order to understand and follow the commands of the scale (mini mental state examination score $\geq 24/30$) [28]. Older adults with conditions that could affect stability during gait (such as recent fractures or lower limb surgeries), or heart and neurologic diseases were excluded from the study. All participants provided their written consent.

#### 2.2.2. Measures

The adapted Greek Functional Gait Assessment scale ($FGA_{GR}$) was performed to test its psychometric characteristics. The ten activities of walking that it contains in various conditions (in changing speed, with head turns, with eyes closed, in backwards direction, with a narrow base of support, with pivot turns, over obstacles, and on stairs), are scored in a 0–3 ordinal scale of a gradually increased performance from 0 ("severe impairment"), 1 ("moderate impairment"), 2 ("mild impairment"), to 3 ("normal"). The maximum score is 30 points [14].

The Berg Balance Scale (BBS) is a well-established balance test with 14 items evaluating stability in various everyday conditions such as from sitting to standing, in standing position, in one leg stance, with eyes closed, during reaching forward and picking up an object from the floor, placing alternate foot on stool, looking over the shoulders, and turning 360° [29]. The items are scored in an ordinal scale (0–4) yielding a total score which ranges from 0 to 56 with higher scores indicating better performance and greater independence [30]. The scale was used for the concurrent validity evaluation of the pre-final FGA$_{GR}$ scale.

The mini-Balance Evaluation Systems Test (mini-BESTest) comprises 14 items evaluating balance systems (anticipatory adjustments, reactive control, compensatory stepping corrections, sensory orientation, dynamic balance and gait) during functional tasks which take only 15 min to be delivered [31]. This balance scale has been recently developed, but its excellent reliability and its strong correlation with other balance measures make it one of the best and most common choices for assessing balance in older adults [32,33].

The Timed Up and Go (TUG) test is a very easy balance test which requires the participant to stand up from a chair, walk 3 m, turn, walk back, and sit down on the same chair [34]. The time to administer the test has been positively associated with a past history of falls, and therefore the TUG is commonly used to predict the risk of falling in elderly people. Although various values have been proposed as cutoff scores, the score of 11.1 min has been found not only to predict future falls, but also to be 80% sensitive and 56% specific in identifying actual falls reported in older adults (over 60 years) with vestibular dysfunction [35].

The Falls Efficacy Scale-International (FES-I) questionnaire was used to explore the participants' "concern" of falling in everyday living activities [36]. It contains 16 questions concerning activities inside and outside the home, it is very easy and quick to complete, and has provided high certainty of evidence supporting its reliability, construct validity, and responsiveness [37]. Each item of the FES-I can be scored from 1 (not at all concerned) to 4 (very concerned), yielding a total score which ranges from 16 (absence of concern) to 64 (extreme concern) [38]. Score > 23 for the scale indicates a high concern about falling [39].

### 2.2.3. Assessors

All measures were administered by the same assessor who was a qualified, experienced physiotherapist and was trained in the use of the scales by the supervisor of the research (SL), who is a neurological physiotherapist with great expertise in using functional scales. Additional help in using the scales was taken by training videos and manuals (https://www.bestest.us/). A second assessor (SL) administered the scale for the inter-rater reliability assessment.

### 2.2.4. Reliability Testing of the FGA$_{GR}$

In order to assess the test-retest reliability, the FGA$_{GR}$ was administered twice by the same assessor within a one week interval, so that the conditions remain the same to the greatest extent possible. The participants were also informed about the same clothing, as well as the same time and the same place that should be kept in both visits. During the first meeting, a second rater also evaluated the participants and filled in the FGA scoring sheet for the inter-rater reliability assessment. The internal consistency reliability, which measures the degree that the items of the scale are all measuring the same construct and are relative to their sum score was also evaluated [27]. In addition, the Minimum Detectable Change at 95% of confidence interval (MDC95%) was tested to evaluate the smallest change in score that reflects a true change in the gait ability of the participant. The amount of variability in the test scores was assessed by the Standard Error of Measurement (SEM) based on the reliability of the scale and the standard deviation of the population [40].

### 2.2.5. Validity Testing of the FGA$_{GR}$

Concurrent validity was assessed by correlating the FGA$_{GR}$ with other measures that assess dynamic balance such as the TUG test, the BBS, and the mini-BESTest. A self-report measure, the FES-I, was also chosen for correlation with the FGA$_{GR}$ to assess the convergent validity of the translated scale. Discriminant validity was assessed via the ability of the scale to distinguish between (i) those who had low concern about falls and (ii) those who reported high concern about falls as this was indicated by the FES-I questionnaire cutoff point of 23 (FES-I $\geq$ 23 high concern) [39]. The predictive validity of the FGA in identifying older adults at risk for falls was also assessed using the TUG cutoff score of 11.1 s, which is considered to be the most appropriate to classify older adults who are at risk for falls [16,35].

### 2.3. Experimental Settings and Procedure

On the appointed day, prior to balance and gait assessments, basic information of each participant was recorded, which included the sociodemographic and clinical characteristics. Following this introductory session, the scales were then administered starting with the BBS and mini-BESTest. To give time for some rest, the FES-I questionnaire was the next measure followed by the FGA. After completing each scale, the participant had a 5 to 10 min break to rest.

Participants completed the FGA in a large room with a delimited 6 m walkway and width markings as directed in the FGA instructions. For each activity, first verbal instructions were given and then, if necessary, a demonstration of the activities was followed. Throughout the evaluation, the evaluators were close to the participant, for his safety. The commands were given by reading the exact instructions text from the scale, to test the clarity of the translation. Any difficulties in understanding the commands or performing the tasks were recorded and if difficulties were present in more than 20% of the cases this was an indication that the translation of the item needed to be revised [27].

### 2.4. Data Analysis

### 2.4.1. Sample Size Calculations

Studies reported excellent test-rest and inter-reliability of the FGA with intra-class correlation coefficient above 0.90 [22,25], which was also expected in the present study. Thus, for the reliability analysis, an alpha level of 0.05 and correlation coefficient above 0.9 were taken into account. Based on sample sizes for reliability analysis between two sets of data (i.e., two judges or two measurements in time), a sample of 21 patients is adequate at an alpha level of 0.05 and correlation coefficient above 0.9 [41].

For validity analysis, the literature reports a high and significant correlation between the FGA and BBS (r = 0.80; $p < 0.001$) [26], (r = 0.84, $p < 0.001$) as well as between the FGA and TUG (r = 0.84, $p < 0.001$) in older adults [16]. Thus, a large effect size was expected for the concurrent and convergent validity. As a result, using the values of 0.05 for the alpha level (2-tailed), 0.85 for the power, and 0.6 for a large effect size, the calculation of the sample size yielded 19 participants for the validity analysis [42].

Considering both the validity and reliability power analysis which yielded a rate between 19 and 21 participants, a 15% attrition rate was estimated for the upper limit, yielding a sample of 24 individuals.

### 2.4.2. Statistical Analysis

Relative reliability was assessed by computing the consistency of the two measurements either between the two repeated assessments in time (test-retest reliability) or between the assessments of the two raters (inter-rater reliability) using the Intraclass Correlation Coefficient (ICC$_{2,2}$) and values <0.5 to indicate poor reliability, 0.51–0.75 moderate reliability, 0.76 to 0.90 good reliability, and >0.90 excellent reliability [43]. For the reliability of the single ordinally scaled items, the weighted kappa coefficient was used with the results to be reported as: <0.0–0.20 "none"; 0.21–0.39 "minimal"; 0.40–0.59 "weak"; 0.60–0.79 "moderate"; 0.80–0.90 "strong"; >0.90 "almost perfect" [44]. The internal consistency reliability was

measured with Cronbach's alpha coefficient with "accepted value" of 0.70, "good" internal consistency at values between 0.70 and 0.80, and "excellent" internal consistency at values above 0.80 [45]. The single-to-total correlation was used to find how every single item of the scale correlated well with the total $FGA_{GR}$ score. Correlations below 0.30 indicated a low degree of correlation between the item and the remaining items [46].

The Minimal Detectable Change with a 95% confidence threshold ($MDC_{95}$) indicates the smallest change that can be interpreted as a real change. Additionally, it is greater than the expected measurement error and it was computed according to the following formula: $MDC_{95} = 1.96 \times SEM \times \sqrt{2}$ [47].

The Standard Error Measurement (SEM) was calculated according to the following formula: $SEM = SD\sqrt{(1 - ICC)}$, where ICC is the coefficient of the test-retest reliability and SD is the standard deviation of the $FGA_{GR}$ total score [48].

Ceiling and floor effects were considered as the percentage score of more than 20% of the participants at the highest and lowest score, accordingly [33].

Concurrent validity was investigated using Pearson's correlation coefficient (rs) for normally distributed data, with values between 0.1 and 0.29 interpreted as "weak association", 0.30–0.49 "low association", 0.50–0.69 "moderate association", 0.70–0.89 "strong association", and 0.90–1.00 "perfect association" [49].

The discriminant validity was calculated with the *t*-test for independent samples between the two groups of low and high concern about falls.

For the ability of the $FGA_{GR}$ scale to classify older adults with high risk for falls, a Receiver Operator Characteristic (ROC) curve was used. The TUG cutoff score of 11.1 s was considered as the reference variable, which dichotomized the sample into those with <11.1 s: without risk of falling ("0") and those with ≥11.1 s: with risk of falling ("1"). The data point closest to the upper left corner of the curve, which is the value that is balanced between high Sensitivity (Sn) and high Specificity (Sp) was chosen as the optimal cutoff score for prediction, and the Area Under the Curve (AUC) to reflect the measure's accuracy. An AUC > 0.7 was considered sufficient for discrimination [50]. To determine whether the selected cutoff score could correctly classify falls risk in older adults, the percentage accuracy of the older adults who actually were at risk of falling was calculated using the cutoff score. The Sn, Sp, Positive Likelihood Ratio (LR+), Negative Likelihood Ratio (LR−), Positive Predictive Value (PPV), and Negative Predictive Value (NPV) were also calculated. Sensitivity (Sn) was defined as the proportion of individuals with risk of falling who have a positive result on $FGA_{GR}$, while specificity (Sp) was defined as the proportion of individuals without risk of falling who have a negative result on $FGA_{GR}$ [51]. Positive likelihood ratio (LR+) was considered as the ratio of probability that a person with the risk of falling tested positive to the probability that a person without the risk of falling tested positive (LR+ = true positive/false positive). Negative likelihood ratio (LR−) was considered as the ratio of probability that a person with the risk of falling tested negative to the probability that a person without the risk of falling tested negative (LR− = false negative/true negative) [52]. The PPV indicated the probability that participants with a positive $FGA_{GR}$ result are indeed at high risk of falling while the NPV indicated the probability that participants with a negative $FGA_{GR}$ result are indeed not at high risk of falling [53].

All data were presented as mean ± standard deviation (mean ± SD), and statistical significance was set at $p \leq 0.05$. Statistical analysis was performed with SPSS (version 28.01, SPSS for Windows, SPSS Inc., Chicago, IL, USA).

## 3. Results

### 3.1. Translation and Adaptation Process

All items were characterized as easy to be translated by all translators yielding a range of grades between "1" and "2" in regard to the ease of translation process. The only change that was made compared to the original scale was the absence of the "feet" measure in all dimensions. Given that the official measure for distances in the target language is in meters,

all dimensional characteristics (i.e., the length and width of the walkway) were presented in meters and centimeters.

Regarding the clarity of the commands and the comprehensiveness of the instructions, these were characterized as clear and easy to follow by both therapists and participants. No item was needed to be changed in its translation, and thus the final version of the FGA$_{GR}$ was proceeded for further psychometric evaluation.

### 3.2. Psychometric Testing of Final FGA$_{GR}$

Twenty-four [24] older adults (six men, eighteen women), of mean age 74 $\pm$ 6 years, participated in the psychometric testing of the pre-final FGA$_{GR}$ scale. Full characteristics description is reported in Table 1. Due to the unequal distribution of men and women in the group, a separate analysis with "gender" as a between groups variable was undertaken. No significant differences were revealed in any of the main measures between men and women as shown in Table 2.

**Table 1.** Sample characteristics.

| Characteristics | Mean (Standard Deviation) |
| --- | --- |
| Total Sample | n = 24 |
| Age (y) | 74 (7) |
| Range (min-max) | 66–94 |
| Sex (m/f) | 6/18 |
| Height (cm) | 161 (7) |
| Weight (kg) | 74 (13) |
| Body Mass Index (kg/cm$^2$) | 28 (4) |
| Mini-Mental State (/30) | 28 (1) |

**Table 2.** Characteristics of the sample between men and women at baseline (significant differences between groups at value $p \leq 0.05$).

| Characteristics | Men Mean (Standard Deviation) | Women Mean (Standard Deviation) | *p* Value |
| --- | --- | --- | --- |
| Age (years) | 72 (5) | 80 (9) | 0.093 |
| Body Mass Index (kg/cm$^2$) | 27 (2) | 29 (5) | 0.560 |
| FGA$_{GR}$ total score (/30) | 25 (3) | 28 (3) | 0.064 |
| Mini-BESTest total score (/28) | 20 (3) | 18 (3) | 0.167 |
| TUG (seconds) | 13.3 (3.4) | 13.5 (2.8) | 0.891 |
| FES-I total score (/64) | 24 (5) | 26 (4) | 0.297 |

#### 3.2.1. Reliability

The scale yielded an excellent inter-rater (ICC = 0.984; 95% CI 0.963–0.993; SEM = 0.459, $p < 0.001$) and test-retest reliability (ICC = 0.976; 95% CI 0.945–0.989; SEM = 0.564, $p < 0.001$). Single items agreement between raters and repeated measures was strong to almost perfect for all items (weighted kappa above 0.75). Only items 6 and 7 presented moderate inter-rater agreement and weak to moderate test-retest agreement (Table 3). The scale demonstrated low internal consistency (Cronbach's alpha = 0.66) with the item to total correlation to be statistically significant for all, except for items 1 ("gait level surface"), 6 ("step over obstacle"), and 10 ("steps"), which yielded the weakest correlations with total FGA score (Table 3).

**Table 3.** Single items agreement between raters and repeated measures as this was measured via the weighted kappa coefficient from n = 24 older adults.

| Item of $FGA_{GR}$ | Single Items Agreement | | Item/Total Correlation |
|---|---|---|---|
| | Inter-Rater | Test-Retest | |
| 1 | 0.879 * | 0.818 * | 0.275 |
| 2 | 0.875 * | 0.871 * | 0.490 * |
| 3 | 0.903 * | 0.950 * | 0.485 * |
| 4 | 0.943 * | 1.000 * | 0.732 * |
| 5 | 0.833 * | 0.667 * | 0.575 * |
| 6 | 0.632 * | 0.514 * | 0.339 |
| 7 | 0.703 * | 0.652 * | 0.619 * |
| 8 | 0.936 * | 0.749 * | 0.524 * |
| 9 | 0.849 * | 1.000 * | 0.561 * |
| 10 | 1.000 * | 0.941 * | 0.375 |

* $p < 0.05$.

### 3.2.2. $MDC_{95}$, SEM, Distribution of the Scores

An MDC at 95% confidence interval of 1.56 points on the scale was yielded, with SEM of 0.564.

No ceiling or floor effect was revealed as 0% of the participants presented the highest or the lowest total score on $FGA_{GR}$, respectively.

### 3.2.3. Validity

In regards to the concurrent validity, the pre-final $FGA_{GR}$ scale presented a strong correlation with the mini-BESTest (rs = 0.813, $p < 0.001$), the BBS (rs = 0.748, $p < 0.001$), and a moderate association with the TUG (rs = 0.606, $p < 0.05$).

The convergent validity was presented as low with an expected negative association with the FES-I (rs = 0.365, $p > 0.05$).

The scale also presented a discriminant ability to classify individuals who had high concern about falls (sixteen participants, FES-I mean total score 21.8 ± 3.7) from those with low concern (eight participants, FES-I mean total score 24.5 ± 2.9) ($t_{(22)}$ = 1.783, $p = 0.04$).

The ROC curve results were obtained from 22 participants (19 with risk for future falls and 3 without risk for falls). The ROC curve for the $FGA_{GR}$ showed an AUC of 0.921 (95% CI 0.80–1.00; SEM = 0.063, $p = 0.022$), indicating the ability of the scale to identify individuals at risk for future falls (Figure 1).

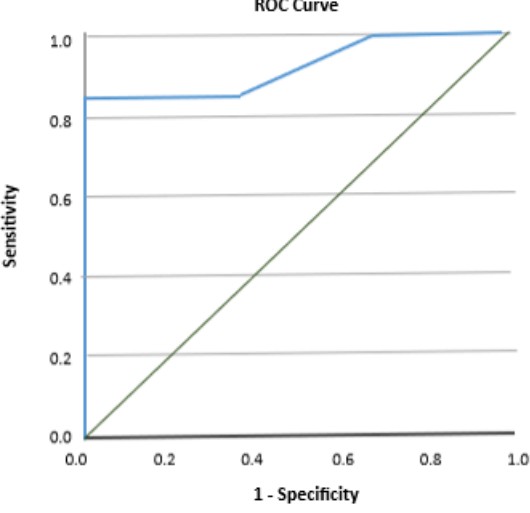

**Figure 1.** Receiver operating curve (ROC) for identifying participants with risk for falls using the scores from the Greek functional gait assessment scale based on the "Timed Up and Go" (TUG) test (scores ≥ 11 s indicated an increased risk for falls).

The cutoff score that balances between high sensitivity and high specificity for the $FGA_{GR}$ was ≤25/30 which resulted in Sn of 84% and Sp of 100%. Taking this cutoff point into consideration, the $FGA_{GR}$ was able to identify 16/19 individuals with risk for falls and all 3 without risk for falls. All predictive values for the $FGA_{GR}$ are listed in detail in Table 4.

**Table 4.** Ability of the $FGA_{GR}$ to identify participants with high risk for falls.

| Variable | $FGA_{GR}$ |
|---|---|
| AUC | 0.921 (0.80–1.00) |
| Cutoff point (/30) | 25 |
| Sensitivity (95% CI) | 0.84 (0.64–0.96) |
| Specificity (95% CI) | 0.50 (0.16–0.84) |
| True Positive | 16 |
| False Positive | 3 |
| True Negative | 3 |
| False Negative | 0 |
| Positive Likelihood Ratio (LR+) | 0 |
| Negative Likelihood Ratio (LR−) | 0.16 |
| Positive Predictive Value (%) | 100 |
| Negative Predictive Value (%) | 86 |

## 4. Discussion

The aim of this study was to cross-culturally adapt the FGA scale into the Greek language and to test its psychometric characteristics. Overall, the translation process was considered easy to be performed and the Greek version of the FGA scale was characterized as easy to administer and comprehensible in its content and commands to the patients. No items needed to be changed from the original scale during the translation process, apart from the measurements in inches and feet that were replaced with the equivalent values in centimeters which are used in Greece. The psychometric evaluation revealed a measurement tool that is reliable and valid in assessing the functional gait and able to predict those healthy older individuals who are at risk for falls. The above results provide evidence for the applicability of the adapted $FGA_{GR}$ instrument in a clinical setting.

Indeed, the easy clinical administration of the scale as well as the excellent reliability of the Greek version in repeated measures and within raters support its utility for assessing functional gait in older adults, with the prerequisite that clinicians should be trained before the use of the scale. In the present study, measurements were made by physical therapists with experience in patient assessment and after being trained by the person in charge of the research, who is specialized in the use of functional neurological evaluation instruments. The Brazilian cross-cultural adaptation of the scale as well as the research by Walker et al. are in agreement with the present study, presenting almost perfect inter-rater and test-retest reliability (ICC > 0.90) when the scale was undertaken by community-living older adults [15,25]. The German translation, made in stroke patients, also found excellent reliability (ICC = 0.94) [23]. Although not in healthy elderly, other studies in neurological patients such as patients with Parkinson's disease [54] and people with stroke [55] found similarly high test-retest and inter-rater reliability. Research on patients with vestibular disorders found the lowest inter-examiner reliability of the aforementioned studies (ICC = 0.84) [14].

Despite the almost perfect inter-rater reliability of total scores, the $FGA_{GR}$ presented some variability in single items reliability, with items 6 ("over obstacles") and 7 ("tandem walking") presenting the lowest inter-rater reliability. Similar low single item values for inter-rater reliability were present in the original scale, as well [14]. The difficulty in performing these tasks by the patients may lead the raters to be somehow strict in assigning the normal score to a patient unless he is almost perfect in his performance. This may have led to the variability in scores between raters. The above tasks as well as item 5 ("pivot turn") yielded a similar variability in single item test-retest reliability, indicating again the difficulty in performing the task, and thus the different ways of coping with these items between the repeated measures. The original scale also presented such test-

retest variability in difficult single items [14]. The items that also showed a weak and not significant correlation with the total score are those items that have been previously reported to deteriorate with age such as walking on an even surface (item 1), over obstacles (item 6), and on stairs (item 10). Walker et al. in their functional gait reference data for healthy individuals at different age levels, found that persons aged 70 to 89 years scored lower on item 1, which may be explained by the lower speed that the elderly present [15]. Indeed, Dommershuijsen et al. at a very recent report of gait speed values in Western European community-dwelling older adults, confirmed that gait speed was lower with older age [56]. The most pronounced decline in gait velocity is observed at 71 years, which is very close to the mean age of our sample [57]. Additionally, the decline in walking speed with age has been found to be associated with balance impairment in the elderly [58]. This could explain the low association of items 6 and 10 with the total score. Both of these items require increased stability on one leg stance in order to overcome an obstacle or to step on a stair with the other leg. These balance demanding tasks may reflect a disproportionate execution compared to the overall good performance. The moderate internal consistency for the $FGA_{GR}$ scale is in contrast to the original version of the scale and to the Brazilian adaptation, where the internal consistency was found to be relatively good with Cronbach's alpha coefficient of 0.79 and 0.85, respectively [14,25]. These differences may be explained either to different populations (i.e., the original scale was tested on patients with vestibular problems) or to different sample sizes (the Brazilian adaptation had a bigger sample of 70 elderly individuals). These differences may also be explained by the variability in single items reliability, mentioned above, which indicate inhomogeneity when balance and speed demanding tasks are included.

The $FGA_{GR}$ also yielded moderate to high correlations with the mini-BESTest, BBS, TUG, and FES-I, indicating the concurrent validity of the scale in Greek older adults. This is the first cross-cultural adaptation of the FGA scale that correlates with the mini-BESTest. The mini-BESTest, which emphasizes both static and dynamic balance during walking, has proven to be one of the most valid and reliable scales for assessing balance and has been studied for its validity in various populations [33,59]. It has been highly correlated with other balance and gait assessment instruments, such as the BBS [32] and TUG [60], and can in turn be used to validate other instruments such as the FGA scale. The high correlation of the $FGA_{GR}$ with a scale considered to be one of the most valid in dynamic balance assessment, such as the mini-BESTest, provides evidence for the validity of the adapted scale. It is, of course, necessary to correlate it with other scales, which purely evaluate walking before clear conclusions can be drawn. The low correlation of the $FGA_{GR}$ scale with the FES-I concern of falling questionnaire cannot be compared with similar studies because there is no research that purely correlates the FGA with the FES-I. This may be because the FGA scale is a fairly new assessment tool in the clinical arena. Nevertheless, there are studies that compare the FES-I with other scales such as the BBS where the results showed a negative correlation as in the present study [61,62]. The negative correlation is justified because the better balance a person has, the lower the concern about falling. The moderate correlation may be explained by the diversity of the scales, as one records the patient's subjective perception of his balance and fear of falling, while the other is an objective means of recording performance in balanced gait activities.

In addition, the $FGA_{GR}$ yielded a predictive validity in identifying the risk of falling at a cutoff score of 25/30. This cutoff score presented a sensitivity of 84% and specificity of 100%. Taking this cutoff point into consideration in our study, the $FGA_{GR}$ identified 16/19 individuals with risk for falls and all 3 without risk for falls. This cutoff score is different from the 22/30 proposed by Wrisley et al. and Marques et al. when the scale was administered to elderly people [16,61]. Our study presented larger specificity compared to other studies and had no false positive cases, which is more important than having false negative cases [16]. The differences with the other studies may be due to different assessment tools used for the ROC analysis. In the study of Marques et al., the discrimination between fallers and non-fallers was used, while in the study of Wrisley et al.,

the TUG score of 11 s, classified as having increased risk for falls, was used [16,26]. However, both studies present the same area under the curve (AUC) of 0.92 with our study indicating the ability of both the original and the Greek scale to identify individuals with high risk for future falls. This ability of the scale is another benefit in the falls prevention campaign because providing early intervention may prevent the complications from falls in the elderly, such as morbidity and mortality, reduced functionality, as well as mental and emotional negative effects [63].

In addition, the absence of the scale to exhibit a ceiling or floor effect in this population as well as its low standard error in measurements support the utility of the FGA$_{GR}$ scale in assessing balance and postural stability during gait. The mean detectable change of 1.56 points could be another indicator for improvement of the functional status of a patient following a rehabilitation program which has a clinical value and importance. The literature lacks such information; therefore, no comparisons with other studies can be carried out. However, the ability of the scale to identify changes in the walking ability of older adults as well as its discriminant ability to classify individuals who had high concern about falls are both excellent characteristics of the scale. The fear of falling usually prevents older adults from participating in favorite activities and hobbies, regardless of whether they fall or not [64]. Given its high prevalence and harmful consequences [65], having a scale which could identify those individuals with high concern for falls, would lead clinicians to modify their rehabilitation program in order to improve balance and confidence in the elderly.

The major strengths of the present study were the establishment of the reliability and validity of the scale as well as the cutoff score of identifying individuals with high risk of falling and the minimum detectable change in the score of the scale. However, this study is not without limitations. A disproportionate sample in regards to the size of women compared to men could be a limitation of the study. Due to this difference, we made further analysis to reassure that the gender did not have any effect on our results. Our results did agree with the reports of Walker et al. as well with the results of Lusardi et al., who both found that gender is not a significant predictor of functional performance and that it does not cause differences in performance at any age tested [15,66]. The lack of a third rater mentioned in several corresponding studies [54,67] could be considered as another limitation. A third party evaluation through videotaped sessions could enhance the results; however, the use of videotaped activities does not always convey the correct dimensions and levels of exercise performance, and thus distorting the image of the true performance of an activity.

## 5. Conclusions

In conclusion, the results of the present study support the easy utility of the scale and its comprehensible administration. The inter-rater and test-retest reliability of the FGA$_{GR}$ scale was well established through this study. The scale correlates with most of the means with which it was compared, with most of the results being in agreement with the other studies. The ability of the scale to discriminate between older adults with high concern about falls from those with low concern, as well as a cutoff point of 25/30 to identify individuals with high risk for falls, were confirmed. A mean detectable change of 1.56 points could be further considered as a way to recognize changes in the functional ability of the elderly following a rehabilitation program. The establishment of a clinical importance change would be of further clinical value. Additionally, it would be interesting to test the reliability and validity of the FGA$_{GR}$ in people with a neurological condition, such as Parkinson's disease, multiple sclerosis, stroke, or vestibular disorders.

**Author Contributions:** Conceptualization, S.L. and V.S.; methodology, S.L.; validation, S.L. and A.K.; formal analysis, S.L.; investigation, V.S. and S.L.; resources, S.L. and I.A.G.; writing—original draft preparation, S.L.; writing—review and editing, V.S. and S.L.; visualization, S.L. and A.K.; supervision, V.S. and S.L.; project administration, A.K., P.N., D.G.K., I.A.G. and S.L. All authors have read and agreed to the published version of the manuscript.

**Funding:** This research received no external funding.

**Institutional Review Board Statement:** The study was conducted according to the guidelines of the Declaration of Helsinki, and ethics approval was obtained by the Cyprus National Bioethics Committee (protocol code EEBK/EP/2017/46 and date of approval: 28 March 2018).

**Informed Consent Statement:** Informed consent was obtained from all subjects involved in the study.

**Data Availability Statement:** Restrictions apply to the availability of these data. Data are partly available from the corresponding author with the permission of Cyprus National Bioethics Committee. The data are not publicly available due to ethical restrictions.

**Conflicts of Interest:** The authors declare that the research was conducted in the absence of any commercial or financial relationships that could be construed as a potential conflict of interest.

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
