# Peer review of "Cross-Cultural Adaptation and Psychometric Characteristics of the Greek Functional Gait Assessment Scale in Healthy Community-Dwelling Older Adults"

_applsci, doi:10.3390/app14020520_

Round 1

Reviewer 1 Report

Comments and Suggestions for Authors

Dear authors, thank you for this work on an interesting topic. The study is of interest and seems to has been well performed. I have only minor corrections to propose.

Abstract

Line 18: Full expression needed before the first occurrence of FGA.

Introduction

Too long ! Authors should be more focused on their topic; a reduction of the length is needed (approximatively 50%).

109-122, more discussion than introduction. Please remove

Method

Complete and well described.

Results and discussion

Adequate concerning the subject and the aims of the study.

References

Ref 74 to 81 are not valid….

Author Response

Dear Reviewer,

Thank you for your comments and suggestions. Please see in the attached file our reply to your comments

Thank you 

Reviewer 2 Report

Comments and Suggestions for Authors

Reviewers' comments to Authors:

The study addresses an important issue by assessing the "Cross Cultural Adaptation and Psychometric Characteristics of the Greek Functional Gait Assessment scale in healthy community-dwelling older adults." This is a very interesting study; however, the authors should consider the following comments:

  1. All abbreviations in the abstract should be explained upon their first use (i.e., FGA).
  2. The introduction is too lengthy; the authors should be more concise.
  3. Authors use both numerical citations in parentheses and cite author names. It is recommended to be consistent in the citation style.
  4. The aim of the introduction is to provide the reader with information summarizing the current state of knowledge and identifying knowledge gaps. The authors present detailed correlation coefficients (r values), but it would be advisable to limit the description to the effect without providing specific numerical values.

In the analysis, the authors report values for 6 men and 18 women and conduct further analyses in Table 2. A significant limitation is the small number of men. Were men and women of the same age?

Reviewer 3 Report

Comments and Suggestions for Authors

The cross-cultural adaptation of the FGA in Greek, according to international guidelines, and its psychometric evaluation is the main aims of this research.

The scale correlates with most of the means with which it was compared, with most of the results being in agreement with the other studies.

The FGAGR scale proves its concurrent and discriminant validity and identifies to be optimal for predicting risk of falls in the elderly.

The good psychometric characteristics of the FGAGR confirm its applicability in assessing gait of Greek speaking older adults.

The approach to prove the effectiveness of the proposed method is clearly presented. In addition to examining the validity of the method, consideration is also being given to cases in which the method does not work properly. I think there is no problem with it as a paper.

Round 2

Reviewer 2 Report

Comments and Suggestions for Authors

Accept in the present form